# Communication about Purchase Desires between Children and Their Parents in Croatia

**Vanesa Varga** [1,*], **Mateja Plenković** [2] and **Marina Merkaš** [1]

1   Department of Psychology, Catholic University of Croatia, 10000 Zagreb, Croatia; marina.merkas@unicath.hr
2   Department of Sociology, Catholic University of Croatia, 10000 Zagreb, Croatia
*   Correspondence: vanesa.varga@unicath.hr

**Abstract:** The main aim of this study is to describe the communication between children and parents about children's desired purchases of items in Croatia. Online focus groups were conducted with children ages 11 to 15, and their parents, using a pre-prepared list of questions. The constant comparative method was applied, and the data were coded thematically, meaning data were organized into groups or codes on the basis of repeating keywords in the transcripts. The analysis shows children mostly ask their parents for clothing items and food. The findings indicate children and parents resolve the purchase decisions based on a few communication themes. Children employ persuasion, bargaining, and negotiation communication to acquire their desired items. As a response, parents employ bargaining and negotiation communication, budgeting and financial communication, usefulness and need communication, and postponed purchase communication. This research contributes to a better understanding of child and parent communication related to child purchase wishes and parent–child communication.

**Keywords:** communication; consumer communication; consumer socialization

## 1. Introduction and Theoretical Background

Communication between parents and children is important for a child's socialization (Mahmudova 2023; Vangelisti 2013). Children's future behaviours and communication styles are partly built on how parents communicate with them. Parent–child communication is influenced by, among other factors, the quality of their relationship, the physical and cognitive development of the child, and shifts in peer engagement and autonomous aspirations. Conflict is common since families and their communication must shift due to the child's developmental changes (Collins and Laursen 2004a, 2004b; Díaz Morales et al. 2023). For example, during adolescence, conflicts arise about different issues, including purchase desires. Children's communication about what they want to buy is important in determining their consumer preferences and choices (Buijzen and Valkenburg 2005, 2008; Naim 2023; Valkenburg and Cantor 2001).

The main aim of this study is to describe the communication between children and parents about children's desired purchases of items in Croatia. However, it is important to state that the results presented in this study are not to be generalized, and the term children in Croatia is a term used to describe the sample presented in this study, which was conducted by a focus group. The theoretical background for this research is the consumer socialization theory.

The consumer socialization theory has been frequently utilized to describe how children acquire consumer skills. Moschis and Churchill (1978) introduced the approach to explain how children develop consumer skills and eventually become consumers. This theory emphasizes that children learn by observing, reinforcing, or imitating the behaviour of others. Three agents are important for this: peers, parents, and the media. Children develop consumer skills by communicating with parents and peers about consumption,

products, brands, and advertising (Mikeska et al. 2017; Milberg et al. 2023; Moschis and Churchill 1978; Sotelo-Duarte and Gónzalez-Cavazos 2023). The role of parents is to educate their children about consumption. Children learn from their parents by observing consumer behaviour (Buijzen and Valkenburg 2003, 2005; Keller and Ruus 2014; Wiese and Kruger 2016). Similarly, peers can influence their interest in specific brands when making purchases. Especially during adolescence, children are often exposed to peer pressure (Huang et al. 2012; Niu 2013). The impact of social media has been the subject of recent studies on children's consumer behaviour. Those cover topics like social media's impact on fashion decisions (Park et al. 2023) and how marketing for unhealthy foods targets children (Ares et al. 2023). Due to children's time with the media, they are an important target group for market research. For example, recently, they have been encouraged to purchase by short video content from TikTok (Wei 2023) and through influencers on social media (Soares and Reis 2023). It is important to state that today, children are often influenced by social media in how they express and behave (Anastasiei et al. 2023; Cipolletta et al. 2020; Ma 2022). Children's consumption styles correspond with the ways they use mobile phones, computers, and the internet (Wilska and Pedrozo 2007). For example, recent research points to the great influence of child influencers who promote unhealthy food and drink brands (Alruwaily et al. 2020). Different developmental levels of children may influence their attention and susceptibility to various influences such as commercial media and peer pressure, which in turn affects their consumption behaviours and values (Chung et al. 2021; Ebenezer 2020; Valkenburg and Cantor 2001). Socialization is often used to understand and explain how children develop cognitive, attitudinal, social, and behavioural outcomes in a social context. Socialization is influenced by various contextual and environmental factors, cultural and societal norms, children's psychological development, social behaviour, and identity formation (Grusec and Hastings 2007). There are many cultural differences in socialization, and cultures are often defined as individualistic or collectivistic (Rose et al. 2002; Singh 2014; Yang et al. 2014). Parents in Western cultures are autonomy-oriented, and parents in most Asian, Latino, African, and rural indigenous societies are relationship-oriented (Triandis 2018). Parents in individualistic cultures such as Australia tend to foster independence in their children, and parents in collectivistic countries such as India prioritize interdependence (Rose et al. 2002). Communication styles can vary between individualistic and collectivist cultures. In individualistic cultures, communication style is more direct. It also places more emphasis on independence and individual achievement. Some common themes are personal needs, desires, and aspirations. In collectivistic cultures, communication style is more indirect, and themes are rather focused on the group and community (Triandis 2018). One study found that children from individualistic cultures are more sensitive to peer influence when compared to children from collectivistic cultures (Sheldon et al. 2017). On the other hand, other research stated that the influence of parents and peers on children's attitudes towards brands is more decisive in cultures with high individualism and power distance (Mishra and Maity 2021). Croatia is often described as a highly collective culture (Sheldon et al. 2017). There are two communication styles between parents and children in the context of consumer behaviour: socio-oriented and concept-oriented. Socio-oriented communication tries to maintain peace in the home and encourage obedience. Socio-oriented parents prioritize obedience and adherence to established boundaries while monitoring and controlling their children's behaviour, learning, and consumption. Concept-oriented communication helps children create their worldviews. Parents encourage children to consider possibilities and participate in open talks about a range of topics while focusing on evaluating alternatives based on their merits. Concept-oriented parents involve children in purchasing decisions, value their opinions, and encourage independent decision-making, even for products not intended for the children themselves (Buijzen and Valkenburg 2005; Caruana and Vassallo 2003; Bristol and Mangleburg 2005; Moschis 1985). Children use different strategies to influence their parents' purchasing decisions. Those include emotional, negotiation, simple pleading, and persuasion strategies. Preschool children tend to rely on simple requests

and emotional appeals, while school-aged children use negotiation and persuasion techniques (Otto and Webley 2016; Thaichon 2017; Palan and Wilkes 1997; Shoham and Dalakas 2006). Children are often involved in the purchasing decisions for food products, especially those high in sugar, and are more likely to use persuasion and negotiation strategies when choosing unhealthy foods (Aluvala and Varkala 2020; Baldassarre et al. 2016; Buijzen and Valkenburg 2008; Van der Heijden et al. 2022). The research regarding Croatian children's purchase desires and their communication with parents is lacking. Justinić and Jagodić (2010) found that children with higher family income derive more pleasure from shopping than adolescents with average or lower family income. In another study, Brdovčak et al. (2018) showed that part of the negative effect of economic pressure on child life satisfaction can be explained by the undesirable effect of economic pressure on self-esteem and hope in adolescents.

## 2. Literature Review

Children can have a significant influence on family purchase decisions (Baldassarre et al. 2016). In Western literature, children have been reported to wield a lot of influence in purchase decisions for products such as snacks, toys, children's wear, and cereals. Children have been observed to influence decisions for family products also, such as holiday/vacations, movies, and eating at particular restaurants or even decision-making for the family to eat out (Kaur and Singh 2006; Senevirathna et al. 2022). According to one study, children exert the most influence on items relevant to them, moderate influence on family activities, and the least influence on consumer durables and costly products (Ekström 2007). Numerous studies have focused on communication between children and parents about money and spending because it can help children's financial knowledge and financial literacy (Jorgensen et al. 2017; Serido et al. 2010; Shim et al. 2010). One study found that both parents and children start financial communication. Children express specific desires, and parents try to educate their children on value, costs, and savings (LeBaron et al. 2020).

Other studies have focused on communication children and parents have in the stores. Therefore, so-called pester power has been well documented. This term is used to describe children pestering and nagging when their requests are denied. Younger children eventually use this strategy to obtain what they want (Binder and Matthes 2023; O'Neill and Buckley 2019). One study found that entertainment, advertisement likability, content credibility, quality information and celebrity endorsement had a significant impact on children's pestering power (Gunardi et al. 2023). Contemporary research has shifted from stores to online shopping. For example, one study analysed parents' strategies for children's purchase influence in online shopping. They found that parents strategize children's influence on online shopping through promising, negotiating, and educating. Each approach has sub-techniques, such as wish lists, half-and-half purchasing, and personal credit cards (Williams and Willick 2023). Furthermore, research often shows developmental differences in children's persuasive tactics (Anitha and Bijuna 2016). One observational study showed four-year-old children tried to gain direct control over their parents, and they did not consider their reactions. Children who were six years old also tried to gain control over the parents, but they considered their objections. Their communication was competitive, and the children bargained. Children who were eight years old tried to overcome the potential conflict and to orient the exchange towards cooperation (Axia 1996). Another experimental study with preschool children found that bargains and guarantees were children's most frequently applied strategies. As children became older, they increased their use of positive sanctions (such as offers, bargains, and politeness) and reduced dependence on assertion (such as forceful assertion) (Weiss and Sachs 1991). Regarding how children talk to their parents, research usually points to less emotional tactics and more rational tactics, such as explaining why the product would benefit them (Binder et al. 2021; Shoham and Dalakas 2006). For example, adolescents will value products that are of better quality and/or cool (Grant and Stephen 2005). Similarly, one study showed that children demonstrated a range of sophisticated influence behaviours that included justifying and highlighting the benefits

of purchases, forming coalitions, compromising, and remaining persistent (Thomson et al. 2007). However, one study showed that children exhibit deception when buying certain products, which means they do not tell the whole truth about the product (Bristol and Mangleburg 2005). There is also research that points to cultural differences in communication. One study has found that Chinese children, compared with their Canadian counterparts, use less bilateral influence strategies like reasoning and bargaining but more unilateral influence strategies like playing on emotions and stubborn persuasion (Yang et al. 2014). Another found that children from the non-post-Soviet countries were more likely to apply various influence tactics than their counterparts from the post-Soviet countries (Yen et al. 2023). Recent research reflects the influence of the economy and social status of the family on child–parent communication. This research emphasises the parents' roles in teaching children financial responsibility (Bogenschneider 2024; McDonald and Shum 2024; Screti et al. 2024).

### 3. Current Research

Today, children are exposed to a constant influx of material items through the media and their everyday environment. It is important to understand how children use communication to fulfil their desires. Data for this study are collected using the method of focus groups with children and their parents. For this research, communication themes are defined as a way individuals use language to acquire what they want and acquire solutions (Faerch and Kasper 1984). Consumer behaviour is defined as how consumers decide to buy products (Kardes et al. 2014).

The main aim of this study is to describe the communication between children aged 11 to 15 years and their parents about the children's desired purchases of items in Croatia. The first specific aim is to determine which products children ask their parents to buy. The second specific aim is to determine what communication children employ to acquire their requested purchase items, and the third specific aim is to determine what communication parents employ to answer their children. However, it is important to state that the results presented in this study are not to be generalized, and the term children in Croatia is a term used to describe the sample presented in this study, which was conducted by a focus group.

This research contributes to the growing literature on child–parent communication regarding purchase requests. The research contributes by shedding light on the complexity of parent–child communication related to consumer behaviour. This study is unique, as it provides both children's and parents' perspectives; thus, it contributes to a better understanding of differences and similarities between children and parents when discussing purchase desires. This knowledge can provide better advice to parents when communicating with their children about purchases and potentially prevent conflicts between them. Research regarding children in Croatia and their purchase desires and communication with parents is lacking. This study provides insights into possible cultural differences in communication between parents and children, since previously acquired knowledge in the foreign literature can hardly be applied to understanding a new cultural setting, namely Croatian, before the examination. It is well known that cultural backgrounds influence our way of communicating. We conducted this study to explore possible ways of communicating purchase desires between parents and children. Thus, the study findings cannot be generalized, and we urge future validation of the obtained patterns of communication about purchase desires between children and parents in Croatia.

### 3.1. Design

The research included focus groups with children and parents. The research design included two structured question sets, one for the parents' focus group and the other for the children's. The project's research team designed the list of questions. Members of the research team composed questions according to the aims of a wider project they are involved in. The aim of this project is to examine the role of parents, peers, and media in shaping materialism in children. These wider project data are provided by focus groups

and questionnaires. This paper focuses only on the data from the focus groups related to child–parent communication. The questions for the project were compiled by studying the scientific literature and discussing them with the project members. The basis for questions related to child–parent communication was in the existing literature (Garison et al. 1999; Thaichon 2017).

Regarding the aim of this paper, parents were asked the following questions: Can you describe situations when your child asks you to buy him/her something? What does he/she ask for? What was the communication with the child after his/her request? Can you describe a situation in which a child has asked you to buy him/her something he/she saw in an advertisement or other media content, such as a cartoon or movie? How did you talk to the child about the request? Did you end up buying the item? Can you describe situations where your child asked you to buy him/her something because his/her friends and peers had it? What were those things? How did you respond to those requests? What did your communication look like?

Regarding the aim of this paper, children were asked the following questions: Can you describe the situations in which you shop with your parents? What did you ask them to buy you most often in these situations? Can you describe a situation where you asked your parents to buy you something you saw in an advertisement, cartoon, or movie? How did your parents respond to your request? Can you describe a situation where your parents buy you something you want? How do you talk about it? What are these things?

*3.2. Recruitment*

Recruitment of participants began in February 2023, with e-mails sent to elementary schools in Croatia. The last participant contacted the researchers via e-mail in June 2023. The e-mailed recruitment packet consisted of the following materials: permission from the Ministry of Science and Education of the Republic of Croatia to conduct the research; permission from the Ethics Committee of the Catholic University of Croatia to conduct the research; participant consent for parents and children; and an invitation to participate in the research for school websites. The invitation was also posted on various social media pages of the research project members. In addition, students from the Catholic University of Croatia helped with recruitment through personal contacts. The invitation was sent to parents of children who attend 5th to 8th grade of elementary school in Croatia. The children in those grades are between the ages of 10 and 15 years and can participate in focus groups without the help of adults.

*3.3. Procedure*

Parents who contacted the researchers via e-mail or social media were asked to sign and return participant consent for parents and children. The sample was non-probabilistic and convenient. In the next step, we contacted parents to schedule a time when they and their children would be available to participate in the focus groups. Before the session began, the moderator informed participants about their rights, that the session would be recorded, that the recorded material would be transcribed and destroyed after transcription and that their names would be coded in the transcript (e.g., F1, F2, Mother 1). All participants, including the moderator, had their cameras and microphones turned on during the session. Children and parents had trouble remembering and describing specific communication.

Nine focus groups were conducted with children on Zoom from 22 April to 5 May 2023. Nine trained moderators conducted and transcribed children's focus groups. Nine focus groups with parents were conducted via Zoom from 25 April to 4 June 2023. Five trained moderators conducted and transcribed parents' focus groups. The focus groups had three to six participants.

*3.4. Sample*

The sample consisted of 40 children from 11 to 15 years of age (20 boys; 20 girls; mean age = 14) and 39 parents (4 fathers; 35 mothers). Most children and parents were from Zagreb, the capital city of the Republic of Croatia.

*3.5. Data Analysis*

The constant comparative method was applied, and the data were coded thematically, meaning data were organized into groups or codes on the basis of repeating keywords in the transcripts (Corbin and Strauss 1998; Monalisa et al. 2023). The unit of analysis was a general statement made by a child or a parent. After reading through the transcripts, it was determined that these desires could be categorised into the following codes: Food and Drink (M1), Clothing (M2), Technology (M3), Toys (M4), and Other (M5). A total of 87 statements were coded from children about what they wanted to buy. Coders agreed on a total of 76 statements. The inter-rater reliability for the groups of children's conversations about their purchase requests was 87.35%. A total of 88 statements were coded from parents about their children's purchase desires. Coders agreed on a total of 80 statements. The inter-rater reliability for the parents' discussion groups about their purchase desires was 90%. For the study of parent–child communication related to the purchase of items, the codes were identified after reading through the transcripts and defining repeating themes in statements such as money, usefulness, and others. Children's requests were based on Persuasion communication (C5), and Bargaining and negotiation communication (C1). Parents' responses include the following codes: Bargaining and negotiation communication (C1), Budgeting and financial communication (C2), Usefulness and need communication (C3), and Postponed purchase communication (C4). A total of 49 statements were coded from children. Coders agreed on a total of 45 statements. The inter-rater reliability for the children's discussion groups was 86.53%. A total of 78 parent statements were coded. Coders agreed on a total of 68 statements. The inter-rater reliability for the parent discussion groups was 86.07%.

## 4. Results

*4.1. Purchase Desires of Children*

In this part, we present the results related to the first specific aim: to determine which products children ask their parents to buy, along with a few examples from parents' and children's perspectives. The children's purchase desire most frequently mentioned by children and parents was clothing. Children desire branded products such as Jordan's, Nike, and Adidas. They often indicate wanting the same sneakers or clothes as their friends. Children express a desire for clothes and sneakers that are related to their interests and hobbies. For example, they want appropriate sneakers or clothing if they play sports like basketball. There is also interest in clothing with pictures or logos of popular cartoon characters or music bands they like and clothing they see in media such as television and the internet.

Examples:

F47 (girl, 13): I saw some sneakers that are in fashion now, and my parents bought them for me.

F41 (girl, 11): I once saw a girl who had a cool bag. I found it in a store and showed it to my mom, but it was too expensive, so . . .

Mother 73: He wanted some sneakers he saw on TV, but he knows he has two pairs of new sneakers.

Mother 72: And now they're looking for that Tech Fleece and Adidas sneakers.

The second most common purchase desire of children had to do with various toys. However, these were the ones least mentioned by parents. Boys mentioned toys more, and the most common were Legos. The statements show how diverse children's wishes are

regarding toys (football player trading cards, Lego, etc.). Children often respond to media advertisements for specific products and desire ones like their peers' toys.

Examples:

F11 (boy, 14): Well, when I was watching Spider-Man, because I'm a big fan of Marvel, especially Spider-Man, I asked mom and dad to buy me a Spiderman suit.

F26 (boy, 12): It was the same with me, with Pop-It. Everyone had it, so I told my mom and then she bought it for me.

Mother 20: The other day, they went to the cinema to watch a Marvel movie and now they need Lego figures from that movie.

Mother 32: Those Dormeo owls, that ad had an impact, so we got it.

Food and beverages were the third most frequent purchase items requested by children. However, they ranked as the second item that parents most often cited. Most of the participants agreed that children are drawn to sweets. The most popular kinds of sweets that they ask for include ice cream and candies. Parents said they were willing to treat their children to inexpensive candies. Additionally, parents are open to letting children select the treats they want. That frequently occurs during store trips.

Examples:

F53 (boy, 14): Sweets. But we buy the cheapest. That is what we agreed to.

F53 (boy, 14): Usually, when I go shopping with my mom, I always beg her for sweets and things like that.

Mother 5: Always sweets: "I would like this candy; I would like that candy".

Mother 72: Daily, it is ice cream, especially now that the warmer weather is here, or, I don't know, something from the bakery.

Children's statements on technology were a little less frequent than those by parents. Children desire technology gadgets like mobile phones, laptops, and mostly PlayStation. Some of these goods belong to their classmates, which may impact their desire to fit in.

Examples:

F4 (girl, 14): Yes, a few years ago I wanted to get a PlayStation 4 because almost everyone in my class had one.

F41 (girl, 11): Sometimes I ask for some things in one game I play.

Mother 9: Or PlayStation; ours has a PlayStation 4, and their friend has a PlayStation 5, so they said they would also like that newer version.

Mother 24: *The biggest wish he had, and all his friends already had, was a PlayStation.*

Other requests included a field trip, workout gear, a musical instrument, a pet, literature, and makeup.

Examples:

F30 (girl, 11): I once showed a short video about a puppy, and then I asked my parents if we could maybe get a puppy. A few years later, we got a little dog, and the dog is now two years old.

Mother 29: Now she is already at that stage: "Well, I would like that make-up box, so where can I get it? Look and see if you could order it for me through Ali or something like that".

*4.2. Child–Parent Communication*

4.2.1. How Do Children Communicate Their Purchase Requests to Their Parents?

The second specific aim was to determine what communication children use to make purchase requests. After the initial plea or a question, the child–parent purchase communication unfolds. The dominant communication process children employ is persuasion.

Children mention it more than parents, as it is a crucial way to elaborate on the importance of getting a desired item. However, parents recognize it, too. Children do not use the word persuasion but rather beg and ask for something many times. Sometimes, they try to explain to their parents why a certain thing would benefit them. Some children showed their parents a picture or, in some other way, convinced them that a certain thing was worth buying. Emotional appeals can also influence their parents and create a sense of importance for the desired item.

Examples:

F63 (boy, 12): but then I begged my parents, I showed them that I really wanted to because I watched a lot of those virtual realities. I showed my parents that nothing is terrible, nothing will happen. Then I managed to persuade, like please, please.

F4 (girl, 14): Well, for a long time I was asking them little by little, and finally I got it.

Mother 72: he mentioned that he wanted a mobile phone, he said everyone in the class has it. Then I went to check that and found out that, of course, not everyone has a mobile phone. That is one sentence that was for the purpose of his persuasion.

Mother 73: and then he said: "Well, you know, mom, I dreamed that I was getting those tennis shoes".

Children mention bargaining and negotiation more often. In this way, they can give something back to acquire the item, such as doing chores or giving some money.

Examples:

F58 (boy, 12): I tell my mother, mom, buy it for me, if you buy it for me, I'll do better in school, I'll get better grades, hah, I'll do housework, I'll vacuum the house and everything. Then when she buys, I'll be happy, like in heaven.

Mother 12: A very recent situation, we've been negotiating about underwear for a couple of days now, and my son is in the 7th grade and normally he's not demanding, but he's stubborn, very persistent, so when he sets his mind to something, he tries and tries.

Mother 15: And he says, first, you know, I saw a really great chess set I really like. Then I say—what's wrong with yours, then he says—it's not bad, but his is magnetic, so you know, it is better. So, we negotiate. He has his own money.

4.2.2. How Do Parents Respond to Children's Purchase Requests?

The third specific aim was to determine what communication parents employ to answer their children. The statements below illustrate how parents encourage children to earn rewards through good behaviour or accomplishments. That may include completing chores, getting good grades in school, or participating in activities. Children mention in their statements that they are promised or given gifts for achieving specific goals. Parents often set the conditions for children to obtain the desired things.

Examples:

F64 (girl, 13): But now my mom said I am going to camp on my birthday, so that's like a prize.

F34 (boy, 11): Well, at first, they did not want to buy it for me, so they said they'll give it to me for my birthday. As a present.

Mother 9: If they cannot get something, then we agree that we will get it if they do something well, or that there will be a reward for a good grade.

Mother 18: You must earn it, so we got to the point that everything you want, you must earn it in some way, by working or something.

Budgeting and financial communication were the most frequently mentioned topics in the focus group between parents and children. The focus group emphasised the importance of saving and not buying unnecessarily expensive things. Parents try to find a balance between fulfilling their children's wishes and the financial sustainability of the family. The statements below illustrate that parents talk to their children about financial options and encourage them to understand the value of money. Parents limit the amount of money they spend on a particular product or toy. Children are aware of the family's financial limitations and adjust their wants accordingly. When buying more expensive items, parents often indicate that their children function in a way that saves them money.

Examples:

F53 (boy, 14): I saw a guitar, so I asked my mom for it. So, they told me that they can't afford it now. So, they told us when they get everything we need, they will buy it for me.

F41 (girl,11): I once saw a girl who had a cool bag, so I found it in a store and showed it to my mom, but then it cost a lot, so she said no.

Mother 71: I say it depends on the value of the product, if something is food, a small thing, it's not a problem at all, if it's something larger, then we'll agree when and why we could buy it or not.

Mother 29: they save, and they know very well that some things that are also for them, let's say, some kind of free time, they say "I will buy it with my own money, from my savings.

Parents mention usefulness and need communication more than children, often combined with financial aspects. Parents use conversation to teach their children rational thinking and an understanding of the needs and value of products. They often question whether their children need a particular item and encourage them to consider priorities. Parents have reservations about the requested items' desirability, appropriateness, or practicality. The parents take an active role in evaluating and making decisions about purchasing items for their children. Some parents mention the importance of monitoring trends and considering the cost of the desired item. The conversation highlights the parental role in guiding their children to differentiate between wants and needs.

Examples:

F11 (boy, 14): they were popular slimes before, a year or two ago and this one I saw my friend got that slime, and I really wanted it too, but this one like my mom said it was disgusting and that it wouldn't be very good to have that around the house.

F23 (boy, 11): Well, I was very eager for that toy, and they said they didn't like the way that toy looked, and that was the end.

Mother 22: there is a lot of monitoring of trends, and it comes to us every day with these things, only then is the conversation, do you really need it, how much does it cost.

Mother 14: We then together see if she really needs it or if it is just her wishes that we may not be able to fulfil given our financial possibilities.

Postponed purchase communication was mentioned more in children's focus groups and was sometimes connected to reward, bargaining, negotiation, budgeting, and financial communication. If a child desires something expensive or not immediately necessary, parents agree to provide it in the future after certain conditions are met. The statements highlight that purchasing certain items, such as sneakers or clothes, can be tied to an individual's growth and practicality. The mention of buying something that will be worn longer suggests that the parents consider the longevity and durability of the item before making a purchase. Parents and children also mention specific occasions, such as birthdays, as potential opportunities to acquire desired items. It also suggests a possible strategy for

budgeting and planning purchases around events. The conversation indirectly touches on the concept of delayed gratification.

Examples:

F58 (boy, 12): and if it's for sneakers, then they tell me—we'll buy them in the summer, when my feet grow.

F30 (girl, 11): A few years later, we got a little dog, and the dog is now two years old.

Mother 25: when your feet grow and it will be something you will wear longer, then we will buy it.

Mother 26: so, I say then we'll get it for a birthday, for the New Year, like that.

## 5. Discussion

The first specific aim of this research was to determine which products children in Croatia ask their parents to buy. The statements from children's and parents' focus groups indicate that the most frequently mentioned children's purchase desires are clothing and sugary food. Regarding clothes, children mention the importance of sneakers and sweatshirts. They often name brands and mention peers' influence in forming their purchase desires. That is in accordance with some studies that have highlighted the importance of clothing for children in relation to self-identity, peer group membership, and self-esteem (Badaoui et al. 2012; Chen-Yu and Seock 2002; Wilson and MacGillivray 1998). For example, one study found that child preference for clothing brands is influenced by their identified clothing styles, media, and music. This research also showed that children often request sugary foods like chocolate, sweets, and ice cream. Those are less expensive, so parents often provide them for their children (Chen-Yu and Seock 2002). That is also found in other studies, which showed children's interest in buying food high in sugar (Aluvala and Varkala 2020; Baldassarre et al. 2016).

The second specific aim of this research was to determine what communication children employ to acquire requested purchase items. After the initial request, children employ persuasion tactics such as explaining the item's importance and persistent asking, which is in accordance with previous findings (Caruana and Vassallo 2003; Garison et al. 1999). Both children and parents employ bargaining and negotiation. In this case, the reward is the desired item. Parents use this theme to instil a sense of responsibility and work ethic in their children. By linking rewards to specific achievements or tasks, the parents teach the children the value of putting in effort and working towards their goals. This approach can help develop a sense of responsibility. Rewards can be important in personal growth and development. By setting goals and having rewards tied to them, children are motivated to work harder and improve themselves. That can promote self-discipline and a sense of accomplishment. It is essential to state that both parents and children start this communication. That means that parents tell their children they will buy the item if specific requirements are met, but children also state they will get good grades or do their chores to get the item. The negotiation between children and parents about product purchases has been documented in the literature (Palan and Wilkes 1997; Moschis 1985). The importance of reward to children is also well documented (Badaoui et al. 2012; Wilson and MacGillivray 1998).

Following this, the third specific aim was to determine what communication parents employ to answer their children's requests. The most dominant communication theme revolves around budgeting and finances. Children and parents frequently use the word "expensive" to illustrate the importance of finances in purchasing. Parents initiate this communication. Parents use it to help children learn and discuss priorities, budgeting, and long-term goals. Parents often refer to children saving their own money to acquire expensive items. Through this type of communication and agreement, children learn responsible money management. Parents can carry out the process of financial socialization in an implicit or explicit form (LeBaron and Kelley 2021)—explicit in direct teaching,

recognition of the value of money, distinguishing between needs and wants, and teaching children how to earn, manage and save money (Jazuli and Setiyani 2021), and implicit in a story or game and a role model shown in everyday financial behavior, such as when shopping and saving (Rosalia et al. 2022). The literature emphasises communication with children about money, especially saving their own money (John 1999; Spiro 1983). This communication is often connected to the usefulness of desired things. By evaluating the necessity and practicality of the desired items, the parents aim to teach their children about responsible decision-making, financial literacy, and the value of things. Articles in the literature state that practicality is important to parents who make purchase decisions (Bucciol and Veronesi 2014; Kohls et al. 2009; Towner et al. 2023). Parents also employ postponing purchase communication and connect it to the previous two. This type of communication and agreement is important because it provides delayed gratification and teaches children patience. Studies in the literature state that delayed gratification can benefit children through healthier coping mechanisms and better overall well-being (Horning et al. 2017; Kumar and Pareek 2018; Trzcińska and Sekścińska 2016; Webley and Nyhus 2006). Existing research shows that more nurturing and permissive parental styles, in contrast to authoritarian parental styles, are more likely to yield to children's purchase requests. Therefore, in these cases, children tend to have more influence in their parents' purchases for the entire family (Ndou 2023; Sarwar 2016).

In conclusion, children employ persuasive communication strategies. Parents employ budgeting and financial communication, usefulness and need communication, and postponed purchase communication. And both parents and children employ reward, bargaining, and negotiation. These communication themes show that parents have an important role in shaping their children's consumer behaviour in accordance with the presented consumer behaviour theory. Their strategies teach children the value of other priorities instead of material items, the passing of trends, saving money, financial responsibility, the difference between wants and needs, delayed gratification, etc.

## 6. Limitations and Future Implications

Although this paper has many strengths, the most significant of which is the inclusion of two perspectives, the child's and the parent's, it also has some limitations. The basic limitation refers to the focus group method applied to a topic that can reproduce socially desirable responses from children and parents. That means that children and parents may not have provided specific information and data regarding the purchasing habits of their families and mutual communication. Another limitation is that sometimes, in the focus groups, children and parents had more difficulty remembering specific situations and conversations, so their answers were sometimes general. Future research should explore these qualitative findings in quantitative research to better understand how parent–child communication is associated with various outcomes. This qualitative approach should include a larger sample. Also, future work may explore how social networks contribute to the communication patterns of children and parents. Things desired by children are promoted on social networks that children use, and online shopping is a new way to fulfil these desires.

## 7. Conclusions

This study showed that children usually come to their parents with requests and express various aspirations influenced by peer pressure and the media. Those are mostly clothing and food. The parents strive to guide their children, instil values related to work and effort, and engage in discussions to determine the necessity, budget, and need for requested items. The data show that parents strive to teach their children financial responsibility, encourage rational thinking, and help them understand the difference between needs and wants. Additionally, parents aim to balance fulfilling their children's wishes and maintaining the family's financial stability. The analysis indicates that child–parent communication regarding purchase requests involves various factors, such as children's

desires and parental financial considerations. It highlights the importance of open dialogue, negotiation, and teaching children about responsible decision-making and financial literacy. Therefore, this research underlines the importance of the consumer socialization theory since it shows parents have an important role in shaping their children's consumer behaviour. This research provided us with both children's and parents' perspectives. It provided insights into possible cultural differences in communication between parents and children since the population in Croatia has not been explored.

**Author Contributions:** Conceptualization, V.V. and M.P.; methodology, V.V., M.P., and M.M.; software, V.V., M.P. and M.M.; validation, V.V., M.P. and M.M.; formal analysis, V.V. and M.P.; investigation, V.V. and M.P.; resources, V.V., M.P., and M.M; data curation, V.V., M.P., and M.M.; writing—original draft preparation, V.V., M.P., and M.M.; writing—review and editing, V.V., M.P., and M.M.; visualization, V.V., M.P., and M.M.; supervision, M.M.; project administration, V.V.; funding acquisition, V.V. All authors have read and agreed to the published version of the manuscript.

**Funding:** This research was funded by Catholic University of Croatia. Project titled "Happiness is not in material things: the role of the media, parents and peers in shaping materialism in children" [(HKS-2023)].

**Institutional Review Board Statement:** The study was conducted in accordance with the Declaration of Helsinki, and approved by the Ethics Committee of the Catholic University of Croatia (protocol code KLASA: 6411-03/22-03/037; URBROJ:498-15-06-22-001, 15.9.2022.) for studies involving humans.

**Informed Consent Statement:** Informed consent was obtained from all subjects involved in the study.

**Data Availability Statement:** Data are unavailable due to privacy or ethical restrictions.

**Conflicts of Interest:** The authors declare no conflicts of interest.

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
