# Peer review of "Communication about Purchase Desires between Children and Their Parents in Croatia"

_socsci, doi:10.3390/socsci13020097_

Round 1
Reviewer 1 Report
Comments and Suggestions for Authors
well done

Author Response
Dear Reviewer,
thank you for your valuable suggestions. We have better explained the link between culture and communication’s style.
Line 69-70; Please, can you better explain the link between “culture” (individualistic or collectivistic) and “communication’s style”?
Communication styles can vary between individualistic and collectivist cultures. In individualistic cultures communication style is more direct. It also makes more emphasis on independence and individual achievement. Some common themes are personal needs, desires, and aspirations. In collectivistic cultures communication style is more indirect, and themes are rather focused on the group and community (Triandis, 2018). One research found that children from individualistic cultures are more sensitive to peer influence when compared to children from collectivistic cultures (Sheldon, 2017). On the other hand, other research stated that the influence of parents and peers on children's attitudes towards brands is more decisive in cultures with high individualism and power distance (Mishra and Maity, 2021). Croatia is often described as a highly collective culture (Sheldon, 2017). T
However, we have difficulty understanding the second suggestion related to the aims of the research; Line 405- Please, can you delete this statement. In the main question, there’s no evidence about this “The first specific aim of this research was to determine which products adolescents in Croatia ask their parents to buy”.Line 418 and 434: Please can you delete the words “second and third.
We kindly ask for more detailed suggestion.
In the section current research we have presented one main objective, and 3 specific ones. We have presented the results, and the discussion around those three specific ones: The main aim of this study is to describe the communication between children ages 11 to 15, and parents about children’s desired purchase of items in Croatia. The first specific aim is to determine which products children ask their parents to buy. The second specific aim is to determine what communication children employ to acquire their purchase requests, and the third specific aim is to determine what communication parents employ to answer to their children:
“The first specific aim of this research was to determine which products adolescents in Croatia ask their parents to buy”.
We have stated also: . However, it is important to state that the results presented in this study are not to be generalized, and the term children in Croatia is a term used to describe the sample presented in this study, which was conducted by a focus group. The theoretical background for this research is the consumer socialization theory
According to his first aim we have found on the basis of analysis and presented in the results:
In this part, we present the results related to the first specific aim: to determine which products children ask their parents to buy, along with a few examples from parents' and children perspectives. The most frequently mentioned children's purchase desire by children and parents is clothing. Children desire branded products such as Jordan's, Nike, and Adidas. They often indicate wanting the same sneakers or clothes as their friends. Children express a desire for clothes and sneakers that are related to their interests and hobbies. For example, they want appropriate sneakers or clothing if they play sports like basketball. There is also interest in clothing with pictures or logos of popular cartoon characters or music bands they like and clothing they see in media such as television and the internet.
Following the results we have discussed them as:
The first specific aim of this research was to determine which products children in Croatia ask their parents to buy. The statements from children's and parents' focus groups indicate that the most frequently mentioned children's purchase desires are clothing and sugary food. Regarding clothes, children mention the importance of sneakers and sweatshirts. They often name brands and mention peers' influence in forming their purchase desires. That is in accordance with some studies that have highlighted the importance of clothing for children in relation to self-identity, peer group membership, and self.
Thank you very much in advance.
Reviewer 2 Report
Comments and Suggestions for Authors
Author Response
Dear Reviewer,
thank you for your valuable suggestions. We have made changes in our paper in accordance to your suggestions. The changes in paper, are marked.
The title only refers to adolescent, but the population includes children. So, this title is not appropriate.
We have changed the title to Communication about Purchase Desires between Children and their Parents in Croatia.
Abstract
The characterization of the type of thematic analysis requires greater detail and reference to the author.
We have detailed the analysis in the abstract. However, according to the layout of the Social Sciences we are not able to provide author in the abstract but have provided it in the data analysis section: The constant comparative method was applied, and the data was coded thematically, meaning data was organized into groups or codes on the basis of repeating keywords in the transcripts (Corbin et a., 1998; Monalisa et al., 2023).
Line 9 Need to clarify the sentence: “children in early and middle adolescence” Line 7
We have decided on the suggestion to eliminate the term adolescents and use only the term children in order to make the paper more clear: Online focus groups were conducted with children ages 11 to 15, and their parents, using a pre-prepared list of questions.
Introduction - Need to define/differentiate children and adolescent, noting that the use of both terms is not always clear, for example: “During adolescence, conflicts arise about different issues, including the adolescent's purchase desires. Children's communication about what they want to buy is important in determining their consumer preferences and choices [4, 28 5, 6].” Line 26-28 OR “This theory emphasizes that children learn by observing, reinforcing, or imitating the behaviour of others. Three agents are important for this: peers, parents, and the media. Adolescents develop consumerskills by communicating with parents and peers about consumption, products, brands, and advertising [7, 8, 9, 10]. The role of parents is to educate their children about consumption. Children learn from their parents by observing” Line 39-42 OR “As children became older, they increased their use of 111 positive sanctions(such as offers, bargains, and politeness) and reduced dependence on 112 assertion (such as forceful assertion) [47]. Regarding how adolescents talk to their parents” Line 111-113 OR How do children ask their parents for their purchase requests? The second specific aim wasto determine what communication children use to acquire purchase requests. After the initial plea or a question, the adolescent-parent purchase communication unfolds. The dominant communication process children employ is persuasion.” Line 302-305 I propose adopting the most comprehensive term that includes children and adolescents.
We have decided on the suggestion to eliminate the term adolescents and use only the term children in order to make the paper more clear.The term children now refers to the sample: Online focus groups were conducted with children ages 11 to 15, and their parents, using a pre-prepared list of questions.
Literature Review Here comes another term that can increase the confusion of concepts: “Younger children” Line 98 Please consider the recommendations in the Introduction.
We have specified the term younger children: Preschool children tend to rely on simple requests and emotional appeals, while school aged children use negotiation and persuasion techniques.
Current Research Need to clarify de population: “early and middle-adolescent children” Line 130 What are the exact ages?
The main aim of this study is to describe the communication between children ages 11 to 15, and parents about children’s desired purchase of items in Croatia.
Discussion
The third specific aim, Line 434, benefits from a discussion at the level of parenting styles.
Existing research shows that more nurturing, and permissive parental style in contrast to authoritarian parental style is more likely to yield to children purchase requests. Therefore, in these cases children tend to have more influence in their parents' purchases for the entire family (Ndou, 2023; Sarwar, 2016).
References In the list of 51 references only 12 are from the last 5 years (2018-2023); references must be updated. Many recent studies are available.
We have included the following recent references:
Bogenschneider, Karen. 2024. How the Economy Influences Families and How Families Influence the Economy. Family Policy Matters, New York: Routledge. 187-203.
Brdovčak, Barbara, Marina Merkaš, and Marija Šakić Velić. 2018. Uloga nade i samopoštovanja u odnosu ekonomskoga pritiska i zadovoljstva životom adolescenata. Društvena istraživanja: časopis za opća društvena pitanja 27.1: 87-108.
Caruana, Albert, and Vassallo, Rosella, 2003. Children’s perception of their influence over purchases: the role of parental communication patterns. Journal of consumer marketing 20(1): 55-66.
Díaz Morales, Juan, F., Escribano, Christina, Puig-Navarro, Yaiza and Jankowski, Kondrad, S. 2023. Factors Underpinning the Shift to Eveningness during Early Adolescence: Pubertal Development and Family Conflicts. Journal of Youth and Adolescence 52(3): 561-569.
Justinić, Jasna, and Gordana Kuterovac Jagodić. 2010. Odjeća (ne) čini adolescenta: samopoimanje i potrošačka uključenost u kupovinu odjeće s markom. Drustvena istrazivanja 105/106: 187-208.
Landwehr, Stefanie, and Hartmann, Monika, 2024. Is it all due to peers? The influence of peers on children's snack purchase decisions. Appetite 192: 107111.
Mahmudova, Nigora Hikmatovna. 2023. Influence of family environment on personal socialization. American Journal of Public Diplomacy and International Studies 1.10: 440-446.
McDonald, Tom, and Shum, Holly. 2024. Smartphones, shopping, and the technomobility of migrant mothers. Women’s Agency and Mobile Communication Under the Radar, 43.
Monalisa, Nazratun N., Frongillo, Edward, A.; Blake, Christine, E., Steck, Susan, E., DiPietro, Robin, B.. 2023. Strategies elementary school children use to influence mothers' food purchasing decisions. Maternal & Child Nutrition: e13539.
Naim, Arshi. Consumer Behavior in Marketing Patterns, Types, Segmentation. 2023. European Journal of Economics, Finance and Business Development 1.1: 1-18.
Ndou, Adam Aifheli. 2023. The Relationship Between Parenting Styles and Parental Financial Socialisation. Finance, Accounting and Business Analysis 5(1): 39-48.
Sarwar, Samiullah. 2016. Influence of parenting style on children's behaviour. Journal of Education and Educational Development 3(2): 222-249.
Screti, Cassandra, Edwards, Katie, and Blissett, Jacqueline. 2024. Understanding family food purchasing behaviour of low-income urban UK families: An analysis of parent capability, opportunity and motivation. Appetite: 107183.
Sotelo-Duarte, Manuel, and Gónzalez-Cavazos, Beatriz. 2023. Young consumers influence by older generations: developing the construct and scale to measure intergenerational brand influence. Young Consumers: 253-271.
Williams, David, E., and Willick, Brooklyn. 2023. Co-shopping and E-commerce: parent’s strategies for children’s purchase influence. Electronic Commerce Research, 1-17.
Wiese, Melanie, and & Liezl-Marié Kruger. 2016. Parental influence on consumer and purchase behaviour of Generation Y. Journal of Consumer Sciences 44: 21-31.
Round 2
Reviewer 2 Report
Comments and Suggestions for Authors
Recommendations were introduced with improvement of the manuscript. However, some references may be more current.
Author Response
Dear Reviewer,
Thank you for your suggestion. We have added more recent references to the paper.
References
Alruwaily, Amaal, Mangold, Chelsea, Greene, Tanay, Arshonsky, Josh, Pameranz, Jennifer, and Bragg, Marie. 2020. Child social media influencers and unhealthy food product placement." Pediatrics 146 (5): e20194057.
Aluvala, Ravi, and Varkala, Mallikarjun. 2020. A Study on Impact of Pester Power on Purchase Behaviour of Select FMCG Products
in Hyderabad. International Journal of Management 11(9): 821-829.
Anastasiei, Bogdan, Dospinescu, Nicoleta, and Dospinescu, Octavian. 2023. Word-of-mouth engagement in online social networks: influence of network centrality and density. Electronics 12(13): 2857.
Anitha, P., and Bijuna C. Mohan. 2016. Influence of family structures on pester power and purchase outcomes-a conceptual framework. Procedia Economics and Finance 37 (2016): 269-275.
Ares, Gaston, De Rosso, Sofia, Mueller, Carina, Philippe, Kaat, Pickard, Abigail, Nicklaus, Sophie, and Varela, Paula. 2023. Development of food literacy in children and adolescents: implications for the design of strategies to promote healthier and more sustainable diets. Nutrition Reviews: 1-17.
Axia, Giovanna. 1996. How to persuade mum to buy a toy. First Language 16(48): 301-317.
Badaoui, Khafid, Lebrun, Anne-Marie, and Bouchet, Patrick. 2012. Clothing style, music, and media influences on adolescents’ brand consumption behavior, Psychology & Marketing 29(8): 568-582.
Baldassarre, Fabrizio, Raffaele Campo, and Amedeo Falcone. 2016. Food for kids: How children influence their parents purchasing decisions." Journal of Food Products Marketing 22.5: 596-609.
Binder, Alice, and Matthes, Jorg. 2023. What can stop the ‘pester power’? A longitudinal study on the impact of children's audiovisual media consumption on media‐motivated food purchase requests. Pediatric Obesity 18(6): e13018.
Binder, Alice, Brigitte Naderer, and Jörg Matthes. 2021. Shaping healthy eating habits in children with persuasive strategies: Toward a typology. Frontiers in Public Health 9: 676127.
Bogenschneider, Karen. 2024. How the Economy Influences Families and How Families Influence the Economy. Family Policy Matters, New York: Routledge. 187-203.
Brdovčak, Barbara, Marina Merkaš, and Marija Šakić Velić. 2018. Uloga nade i samopoštovanja u odnosu ekonomskoga pritiska i zadovoljstva životom adolescenata. Društvena istraživanja: časopis za opća društvena pitanja 27.1: 87-108.
Bristol, Terry, and Mangleburg, Tamara, F. 2005. Not telling the whole story: Teen deception in purchasing. Journal of the Academy of Marketing Science 33(1): 79-95.
Bucciol, Alessandro, and Veronesi, Marcella. 2014. Teaching children to save: What is the best strategy for lifetime savings? Journal of Economic Psychology 45: 1-17.
Buijzen, Moniek., and Valkenburg, Patti, M. 2008. Observing purchase-related parent–child communication in retail environments: A developmental and socialization perspective. Human Communication Research 34(1): 50-69.
Buijzen, Moniek., and Valkenburg, Patti, M. 2005. Parental mediation of undesired advertising effects, Journal of Broadcasting & Electronic Media 49(2): 153-165.
Buijzen, Moniek., and Valkenburg, Patti, M. 2003. The effects of television advertising on materialism, parent–child conflict, and unhappiness: A review of research. Journal of applied developmental psychology 24(4): 437-456.
Carlson, Les, Grossbart, Sanford, and Walsh, Ann. 1990. Mothers' communication orientation and consumer-socialization tendencies. Journal of Advertising 19(3): 27-38.
Caruana, Albert, and Vassallo, Rosella, 2003. Children’s perception of their influence over purchases: the role of parental communication patterns. Journal of consumer marketing 20(1): 55-66.
Chen‐Yu, Jessie, H., Seock, Yoo- Kyoung. 2002. Adolescents' clothing purchase motivations, information sources, and store selection criteria: a comparison of male/female and impulse/nonimpulse shoppers. Family and Consumer Sciences Research Journal 31(1): 50-77.
Cipolletta, Sabrina, Malighetti, Clelia, Cenedese, Chiara, and Spoto, Andrea. 2020. How can adolescents benefit from the use of social networks? The iGeneration on Instagram. International Journal of Environmental Research and Public Health 17(19): 6952.
Collins, Andrew, W., and Laursen, Brett. 2004a. Changing relationships, changing youth: Interpersonal contexts of adolescent development. The Journal of Early Adolescence 24(1): 55-62.
Collins, Andrew, W., and Laursen, Brett. 2004b. Parent‐adolescent relationships and influences, Handbook of adolescent psychology, 331-361.
Corbin, Juliet, and Anselm Strauss. 1998. Techniques and procedures for developing grounded theory. Basics of Qualitative Research. New York: Sage.
Chung, Alicia, Vieira, Dorice, Donley, Tifanny, Tan, Nicholas, Jean-Louis, Girardin, Gouley, Kathleen Kiely, and Seixas, Azizi. 2021. Adolescent peer influence on eating behaviors via social media: scoping review. Journal of medical Internet research 23(6): e19697.
Díaz Morales, Juan, F., Escribano, Christina, Puig-Navarro, Yaiza and Jankowski, Kondrad, S. 2023. Factors Underpinning the Shift to Eveningness during Early Adolescence: Pubertal Development and Family Conflicts. Journal of Youth and Adolescence 52(3): 561-569.
Ebenezer, Odji. 2020. Influencing Children: Limitations of the Computer-Human-Interactive Persuasive Systems in Developing Societies. International Journal of Modern Education & Computer Science 12.5: 1-15.
Ekström, Karin. 2007. Parental consumer learning or “keeping up with the children.” Journal of Consumer Behaviour 6(4): 203–217.
Faerch, Claus, and Kasper, Gabiele. 1984. Two ways of defining communication strategies. Language learning 34(1): 45-63.
Garison, Betsy, M.,. Pierce, Sarah, H., Monroe, Pamela, A., Sasser, Diane, D., Shaffer, Amy, C. and, Blalock, Lydia, B. 1999. Focus group discussions: Three examples from family and consumer science research. Family and Consumer Sciences Research Journal 27(4): 428-450.
Grant, Isabel J., and Graeme R. Stephen. 2005. Buying behaviour of “tweenage” girls and key societal communicating factors influencing their purchasing of fashion clothing. Journal of Fashion Marketing and Management: An International Journal 9.4: 450-467.
Grusec, Joan, E., and Hastings, Paul, D. 2007. Handbook of socialization: Theory and research, New York: The Guliford Press.
Gunardi, Ardi, Swati Mathur, and Eddy Jusuf. 2023. Television Advertisements: Children's Pestering Power Influence on Parents Buying Behaviour." International Journal of Professional Business Review 8.4: e01852-e01852.
Van der Heijden, Amy, te Molder, Hedwig, Huma, Bogdana, and Jager, Gerry. 2022. To like or not to like: negotiating food assessments of children from families with a low socioeconomic position. Appetite 170 (2022): 105853.
Horning, Melissa, L., Fulkerson, Jayne, A., Friend, Sarah, E., and Story, Mary. 2017. Reasons parents buy prepackaged, processed meals: it is more complicated than “I Don't Have Time”. Journal of Nutrition Education and Behavior 49(1): 60-66.
Huang, Yunhui, Wang, Lei, and Shi, Junqui, 2012. How attachment affects the strength of peer influence on adolescent consumer behavior. Psychology & Marketing 29(8): 558-567.
Jazuli, Aroh, and Setiyani, Rediana. 2021. Anteseden Financial Management Behavior: Financial Literacy Sebagai Intervening. Economic Education Analysis Journal 10(1): 163-176.
John, Deborah, 1999. Consumer socialization of children: A retrospective look at twenty-fve years of research. Journal of Consumer Research 26(3). 183-213.
Jorgensen, Bryce, L., Rappleyea, Damon, L., Schweichler, John, T., Fang, Xiangming, Moran, Mary, E. 2017. The financial behavior of emerging adults: A family financial socialization approach. Journal of Family and Economic Issues 38: 57-69.
Justinić, Jasna, and Gordana Kuterovac Jagodić. 2010. Odjeća (ne) čini adolescenta: samopoimanje i potrošačka uključenost u kupovinu odjeće s markom. Drustvena istrazivanja 105/106: 187-208.
Kardes, Frank, Cronley, M.aria, and Cline, Thomas. 2014. Consumer behavior. South-Western Cengage Learning.
Keller, Margit, and Ruus, Riina, 2014. Pre‐schoolers, parents and supermarkets: co‐shopping as a social practice. International Journal of Consumer Studies 38(1): 119-126.
Kaur, Pavleen, and Singh, Raghbir. 2006. Children in Family Purchase Decision Making in India and the west. Academy of Marketing Science Review 8(8).
Kohls, Gregor, Peltzer, Judith, Herpertz‐Dahlmann, Beate, and Konrad, Kerstin. 2009. Differential effects of social and non‐social reward on response inhibition in children and adolescents. Developmental science 12(4): 614-625.
Kumar, Sachin, and Pareek, Kumkum. 2018. Role of ability to delay gratification and regulate emotions in adolescents' psychological well-being. Indian Journal of Positive Psychology 9(2): 215-218.
Landwehr, Stefanie, and Hartmann, Monika, 2024. Is it all due to peers? The influence of peers on children's snack purchase decisions. Appetite 192: 107111.
LeBaron, Ashley, B., Marks, Loren, D., Rosa, C. M., and Hill, Jeffrey. 2020. Can we talk about money? Financial socialization through parent–child financial discussion. Emerging Adulthood, 8(6): 453-463.
LeBaron, Ashley B., and Kelley, Heather H. 2021. Financial Socialization: A Decade in Review. Journal of Family and Economic Issues, 42(S1), 195–206.
Ma, Cecilia, M. 2022. Relationships between social networking sites use and self-esteem: the moderating role of gender. International Journal of Environmental Research and Public Health 19(18): 11462.
Mahmudova, Nigora Hikmatovna. 2023. Influence of family environment on personal socialization. American Journal of Public Diplomacy and International Studies 1.10: 440-446.
McDonald, Tom, and Shum, Holly. 2024. Smartphones, shopping, and the technomobility of migrant mothers. Women’s Agency and Mobile Communication Under the Radar, 43.
Mikeska, Jessica, Harrison, Robert, L., and Carlson, Les. 2017. A meta-analysis of parental style and consumer socialization of children. Journal of Consumer Psychology 27(2): 245-256.
Milberg, Sandra, J., Cuneo, Andres, Silva, Monic, Goodstein, Ronald, C. 2023. Parent brand susceptibility to negative feedback effects from brand extensions: A meta‐analysis of experimental consumer findings. Journal of Consumer Psychology 33(1): 21-44.
Mishra, Anubuhav, and Maity, Moutusy, 2021. Influence of parents, peers, and media on adolescents' consumer knowledge, attitudes, and purchase behavior: A meta‐analysis. Journal of Consumer Behaviour 20(6): 1675-1689.
Monalisa, Nazratun N., Frongillo, Edward, A.; Blake, Christine, E., Steck, Susan, E., DiPietro, Robin, B.. 2023. Strategies elementary school children use to influence mothers' food purchasing decisions. Maternal & Child Nutrition: e13539.
Moschis, George, P. 1985. The role of family communication in consumer socialization of children and adolescents. Journal of consumer research 11(4): 898-913.
Moschis, George, P., and Churchill Jr, Gilbert, A. 1978. Consumer socialization: A theoretical and empirical analysis. Journal of marketing research 15(4): 599-609.
Naim, Arshi. Consumer Behavior in Marketing Patterns, Types, Segmentation. 2023. European Journal of Economics, Finance and Business Development 1.1: 1-18.
Ndou, Adam Aifheli. 2023. The Relationship Between Parenting Styles and Parental Financial Socialisation. Finance, Accounting and Business Analysis 5(1): 39-48.
Niu, Han-Jen, 2013. Cyber peers’ influence for adolescent consumer in decision‐making styles and online purchasing behavior. Journal of Applied Social Psychology 43(6): 1228-1237.
O’Neill, Claire, and Buckley, Joan. 2019. “Mum, did you just leave that tap running?!” The role of positive pester power in prompting sustainable consumption. International Journal of Consumer Studies 43(3): 253-262.
Otto, Annette, and Webley, Paul. 2016. Saving, selling, earning, and negotiating: How adolescents acquire monetary lump sums and who considers saving. Journal of Consumer Affairs 50(2): 342-371.
Palan, Kay, M.; Wilkes, Robert, E. 1997. Adolescent-parent interaction in family decision making. Journal of Consumer Research 24(2): 159-169.
Park, Sohyun, Lee, Seung, Hee, Merlo, Caitlin, and Blanck, Heidi, M. 2003. Associations between Knowledge of Health Risks and Sugar-Sweetened Beverage Intake among US Adolescents. Nutrients 15(10): 2408.
Rosalia, Vicky, Sari Simatupang, Dewi R.atna, and Anggia, Yola. 2022. Improving Financial Literacy Knowledge from An Early Age by Socialization and Training to The Elementary School in Medan City. Jurnal Pengabdian Kepada Masyarakat, 7(2).
Rose, Gregory, M., Dalakas, Vassilis, Kropp, Fredric, 2002. A five-nation study of developmental timetables, reciprocal communication and consumer socialization. Journal of Business Research 55(11): 943-949.
Sarwar, Samiullah. 2016. Influence of parenting style on children's behaviour. Journal of Education and Educational Development 3(2): 222-249.
Screti, Cassandra, Edwards, Katie, and Blissett, Jacqueline. 2024. Understanding family food purchasing behaviour of low-income urban UK families: An analysis of parent capability, opportunity and motivation. Appetite: 107183.
Serido, Joyce, Shim, Soeyon, Mishra, Anubha, Tang, Chuanyi. 2010. Financial parenting, financial coping behaviors, and well-being of emerging adults. Family Relations 59: 453–464.
Senevirathna, S. D., P. Wachissara Thero, and PO De Silva. 2022. A study of children’s influence in family purchasing decisions: parents’ perspective." Asian Journal of Marketing Management 1.01: 47-65.
Sheldon, Pavica, Rauschnabel, Philipp A., Antony, M.ary Grace, and Car, Sandra. 2017. A cross-cultural comparison of Croatian and American social network sites: Exploring cultural differences in motives for Instagram use. Computers in human behavior 75: 643-651.
Shim, Soyeon, Barber, Bonnie, L., Card, Noel, A., Xiao, Jing Jian, and Serido, Joyce. 2010. Financial socialization of first-year college students: The roles of parents, work, and education. Journal of Youth and Adolescence 39: 1457–1470.
Shoham, Aviv, and Dalakas, Vassilis. 2006. How our adolescent children influence us as parents to yield to their purchase requests. Journal of Consumer Marketing 23(6): 344-350.
Singh, Anamika. 2014. A comparative study of individualism vs collectivism and its impact on Indian youth culture with special reference to television commercials. IJIRST–International Journal for Innovative Research in Science & Technology 1(6): 139-151.
Soares, Diana, and Reis, Jose Louis. 2023. Behaviour of the Adolescents and Their Parents in Relation to the Micro-Influencers in Instagram. Marketing and Smart Technologies: Proceedings of ICMarkTec 2: 361-374
Sotelo-Duarte, Manuel, and Gónzalez-Cavazos, Beatriz. 2023. Young consumers influence by older generations: developing the construct and scale to measure intergenerational brand influence. Young Consumers: 253-271.
Spiro, Rosann, L. 1983. Persuasion in Family Decision Making, Journal of Consumer Research 9: 393-402.
Thaichon, Park, 2017. Consumer socialization process: The role of age in children's online shopping behavior. Journal of Retailing and Consumer Service 34: 38-47.
Thomson, Elizabet, Laing, Angus, and McKee, Lorna. 2007. Family purchase decision making: Exploring child influence behaviour. Journal of Consumer Behaviour: An International Research Review 6(4): 182-202.
Towner, Emily; Chierchia, Gabriele, and Blakemore, Sarah-Jayne. 2023. Sensitivity and specificity in affective and social learning in adolescence, 2023, Trends in Cognitive Sciences: 642-655.
Triandis, Harry, C. 2018. Individualism and collectivism. New York: Routledge.
Trzcińska, Agata, and Sekścińska, Katarzyna. 2016. The effects of activating the money concept on perseverance and the preference for delayed gratification in children. Frontiers in psychology 7: 609.
Valkenburg, Patti, M., and Cantor, Joanne. 2001. The development of a child into a consumer. Journal of Applied Developmental Psychology 22(1): 61-72.
Vangelisti, Anita, L. 2013. The Routledge handbook of family communication. New York: Routledge.
Webley, Paul, and Nyhus, Ellen, K. 2006. Parents’ influence on children’s future orientation and saving. Journal of Economic Psychology, 27(1): 140-164.
Wei, Binge. 2023. Research on the Influence of Live Webcast on Teenagers' Consumption Behavior under the New Media Background: Taking Live Webcast in Tik Tok as an Example. International Journal of Education and Humanities, 6(3): 142-146.
Wiese, Melanie, and & Liezl-Marié Kruger. 2016. Parental influence on consumer and purchase behaviour of Generation Y. Journal of Consumer Sciences 44: 21-31.
Weiss, Deborah, and Sachs, Jacqueline. 1991. Persuasive strategies used by preschool children. Discourse Processes, 14(1): 55-72.
Williams, David, E., and Willick, Brooklyn. 2023. Co-shopping and E-commerce: parent’s strategies for children’s purchase influence. Electronic Commerce Research, 1-17.
Wilska Terhi-anna, and Pedrozo, Sueila. 2007. New technology and young people’s consumer identities: A comparative study between Finland and Brazil. Young 15(4): 343–368.
Wilson, Jeannette, and MacGillivray, Maureen, S. 1998. Self‐perceived influences of family, friends, and media on adolescent clothing choice. Family and Consumer Sciences Research Journal 26(4): 425-443.
Yang, Zhiyong, Kim, Chankon, Laroche, Michel, and Lee, Hanjoon. 2014. Parental style and consumer socialization among adolescents: A cross-cultural investigation. Journal of Business Research 67(3): 228-236.
Yen, Wen-Shen, Su, Che-Jen, Lan, Yi-Fang, Mazurek, Marica, Kosmaczewska, Joanna, Švagždienė, Biruta, and Cherenkov, Vitally. 2023. Adolescents’ use of influence tactics with parents in family travel decision making: a cross-societal comparison in Eastern Europe. The Social Science Journal 60(3): 478-490.